# A database of glacier prokaryotic genomes and genes for the Three Poles

Yongqin Liu[1,2,3]*, Songnian Hu[3,4], Tao Yu[1,3,4], Yingfeng Luo[3,4], Zhihao Zhang[2,3], Yuying Chen[2,3], Shunchao Guo[4,5,6], Qinglan Sun[4,5,6], Guomei Fan[4,5,6], Linhuan Wu[4,5,6], Juncai Ma[4,5,6], Keshao Liu[2], Pengfei Liu[1], Junzhi Liu[1], Ruyi Dong[1], Mukan Ji[1]*

[1]Center for Pan-third Pole Environment, Lanzhou University, Lanzhou, China
[2]State Key Laboratory of Tibetan Plateau Earth System, Resources and Environment (TPESRE), Institute of Tibetan Plateau Research, Chinese Academy of Sciences, Beijing, China
[3]University of Chinese Academy of Sciences, Beijing, China
[4]State Key Laboratory of Microbial Resources, Institute of Microbiology, Chinese Academy of Sciences, Beijing, China
[5]Microbial Resource and Big Data Center, Institute of Microbiology, Chinese Academy of Sciences, Beijing, China
[6]Chinese National Microbiology Data Center (NMDC), Beijing, China

*Correspondence to*: Yongqin Liu (yql@lzu.edu.cn), Mukan Ji (jimk@lzu.edu.cn)

**Abstract.** Glaciers cover 10% of Earth's land area and are a pool of carbon and nitrogen for downstream ecosystems. Microbes, including bacteria, fungi, algae, and other microeukaryotes, are the primary inhabitants of glacier ecosystems and are key drivers of carbon and nitrogen transformation. Among them, prokaryotes (including bacteria and archaea) are the most diverse and abundant. Here, we present a dataset on supraglacial bacterial and archaeal (referred to as prokaryotic hereafter) communities across the Antarctic, Arctic, Tibetan Plateau, and other alpine regions. The dataset comprises 2,039 amplicon sequencing data, 999 cultured bacterial genomes, and 208 metagenomes, covering ice, snow, and cryoconite habitats. The dataset contains 64,510 prokaryotic amplicon sequencing phylotypes, with a higher diversity in the Tibetan glaciers than in the Antarctic and Arctic glaciers, which were respectively enriched with Gammaproteobacteria, Bacteroidota, and Alphaproteobacteria. The dataset also contains 62,595,715 unique genes and 4,501 prokaryotic genomes, a 35.5% expansion from previous publications. Genes were annotated for those associated with carbohydrate-active enzymes, nitrogen cycling, methane cycling, antimicrobial resistance, and microbial virulence, revealing the dynamic microbial functions in glacial habitats. This comprehensive dataset provides standardized prokaryotic diversity, taxonomy, community structure, and genetic functions of glacial microbiomes. The data can be leveraged to elucidate ecological principles governing the distribution of prokaryotes, to gain insights into the key functional genes for supraglacial microbiomes, to build mechanistic models, and to identify any potential biohazards for policymakers to make informed decisions regarding climate change. The dataset is available at the Global Glacier Genome and Gene Database (https://nmdc.cn/4gdb/).

## 1 Introduction

Glaciers cover 10% of Earth's land area (Cauvy-Fraunié and Dangles, 2019) and are mainly distributed in the Antarctic, Arctic, and Tibetan Plateau (the Three Poles) (Qiu et al., 2008). Glaciers store approximately three-quarters of Earth's freshwater (Boetius et al., 2015) and are also a pool of carbon and nitrogen. It has been estimated that six Pg of carbon are stored in global glaciers. These carbon may be released into downstream ecosystems with glacier runoff (Hood et al., 2015), influencing key elemental cycling in downstream ecosystems. Before carbons and nitrogen are released, they undergo extensive biological transformation (Guo et al., 2022), primarily microbial-driven. Microbes, including bacteria, fungi, algae, and other microeukaryotes are the main inhabitants of glacier ecosystems (Cauvy-Fraunié and Dangles, 2019), while bacteria are the most abundant and diverse These microorganisms employ strategies to survive the glacial conditions, such as strong UV radiation, low temperature, and low carbon and nitrogen nutrients (Ciccazzo et al., 2016). As microorganisms are the key driver of carbon and nitrogen transformation in glacier ecosystems, knowledge of their biogeography and functions can greatly enhance our understanding of the biogeochemical cycling in glacial ecosystems and aid in predicting the impact of climate change.

The glacier as a habitat is not homogeneous and is divided into supraglacial, englacial, and subglacial ecosystems. Compared with other glacier-related habitats, the microorganisms in supraglacial ecosystems are the most active, due to their exposure to external environment and ambient temperature. Supraglacial ecosystems can be further separated into snow, ice, and cryoconite holes (cylindrical depressions formed by the preferential melting of dark debris into the surface, typically comprising surface water and cryoconite at the bottom) (Cook et al., 2016), each of which has distinct microbial composition (Anesio and Laybourn-Parry, 2012). Algae and Cyanobacteria are the primary producers in supraglacial ecosystems, with other heterotrophic microorganisms participating in the transformation and degradation of endogenous and exogenous nutrients (Hotaling et al., 2017, Anesio et al., 2017). Active metabolism is reported in glacial ecosystems; for instance, cryoconite is a source of methane but a sink of carbon dioxide, with a rate of 4.60 µmol $m^{-2}d^{-1}$ and $-1.77$ µmol $m^{-2}d^{-1}$, respectively (Zhang et al., 2021). Furthermore, organisms with photosynthesis, nitrification, and denitrification functions are also widespread in glacier cryoconite (Cameron et al., 2012; Stibal et al., 2020).

It was estimated that the mean microbial abundance in glacier surface meltwater is $10^4$ cells $mL^{-1}$ (Stevens et al., 2022), this quantity may further increase with the enhanced climate warming (Segawa et al., 2005). Some of these naturally occurring microorganisms are known as emerging contaminants, which are not commonly monitored in the environment but have the potential to enter the environment and cause known or suspected adverse ecological and/or human health effects (Taheran et al., 2018). A previous study cultivated hemolytic bacteria from Spitsbergen glacier meltwater with potential pathogenicity (Mogrovejo-Arias et al., 2020). Other emerging contaminants in glaciers such as antibiotic resistance genes and microbial virulence factors have also received increased attention (Mao et al., 2023).

Here, we present a glacier dataset on glacial prokaryotes (bacteria and archaea) across the Antarctic, Arctic, Tibetan Plateau, and other alpine regions. This dataset includes amplicon sequencing data from 2,039 samples, 999 cultured bacterial genomes,

as well as shotgun metagenomic sequencing from 224 samples (**Fig. 1**). From an ecological perspective, this dataset with standardized prokaryotic diversity, taxonomy, and community structure can improve understanding of the ecological principles governing the distribution of microorganisms across glaciers, as well as their partitioning across the various habitats in the supraglacial ecosystem. From a geochemical cycling perspective, the database can provide insights into the key functional genes for supraglacial microbiomes, which can be used to better comprehend carbon and nitrogen cycling and allow the building of a model to anticipate glacial carbon and nitrogen dynamics in the future. The dataset archives glacial-specific microorganisms and unique genes in digital form, thus representing an invaluable resource for bioprospecting. Additionally, the dataset can be employed to identify any potential biohazards (pathogens and emerging contaminants) of glaciers and evaluate the impact of glacier melting on downstream ecosystems from a biosafety perspective, thereby assisting policymakers in making informed decisions regarding climate change.

## 2 Materials and methods

### 2.1 Data acquisition

**Amplicon data**: Based on the keywords of "glacier" OR "snow" OR "ice" OR "cryoconite" with sample type being DNA and instrument of Illumina, we retrieved 225,378 SRA initially. Then the results were filtered manually to remove non-supra-glacier habitats (such as glacier forefield, subglacial sediment, proglacial lakes, ice cave), metagenome data, and primers that do not amplify the V4 region of the 16S rRNA gene (i.e., those amply V3V4, V4, and V4V5 were retained) (**Table S1** and **S2**).

**Metagenome data**: All articles containing the keyword "glacier metagenome" were retrieved using the Web of Science (searched on the 1st of December 2022). Only studies with sequenced ice, snow, or cryoconite samples with raw sequence data uploaded on the NCBI Short Read Archive were kept. Additionally, a few metagenome data without published articles were added from IMG/M database and NMDC database based on keyword search by terms "ice", "snow" and "cryoconite". In addition to metagenomes from the Antarctic, Arctic, and Tibetan Plateau, metagenomes from the Andes and Alps were also downloaded. (**Table S3**).

**Cultivated bacterial genome data**: 883 isolate genome data of Tibetan Plateau glaciers were obtained from the TG2G dataset (Liu et al., 2022) and other 116 genomes of bacterial isolates from glaciers beyond the Tibetan Plateau were downloaded from the NCBI Genome database based on keyword search by terms "Antarctic", and "Arctic". After careful manual curation, only samples from ice, snow, and cryoconite habitats were kept (**Table S4**).

### 2.2 Amplicon sequencing data processing

Sequencing data were processed using the USEARCH v11 pipeline (Edgar, 2010) on a sequencing project basis. For each NCBI sequencing project, the reads associated with the project were first merged and quality screened with a max expected error threshold of 0.5, while single-end reads were directly quality screened using the same threshold. The quality-filtered

reads from each bioproject were aligned against the SILVA reference alignment (release 128) to ensure that the V4 hypervariable region is covered, using the align.seqs command in Mothur. After removing any sequences that do not cover the V4 region (using screen.seqs in Mothur), the remaining sequences were dereplicated. After this pre-processing was completed for all bioprojects, the dereplicated sequences of different bioproject were combined, and dereplicated again. Then, these further dereplicated sequences were clustered with 97% identity and chimeric sequences were identified and removed using

the cluster_otus command in USEARCH. The representative sequences were used as the references for OTU table construction. Additionally, the phylotype representative sequences were taxonomically classified using the Bayesian classifier against the Silva database (release 132) (Quast et al., 2012). Then mitochondria, chloroplast, and eukaryotic sequences were removed from the OTU table. After removing samples with less than 5000 reads, the final OTU table comprises 2,039 samples and 64,510 OTUs. The sequencing depth (number of reads) ranges from 5036 to 1,492,659 per sample. We retained two datasets,

one without rarefaction (**Table S5**) and another were subsampled (rarefied) to 5036 reads (**Table S6**).

We calculated the Shannon diversity, richness (number of phylotypes), evenness, and Good's Coverage indices for both original and subsampled data using R. The alpha diversity indices (richness, evenness, and Shannon diversity) and the relative abundance of dominant taxonomy lineages were compared using Kruskal-Wallis one-way ANOVA by region (Antarctic, Arctic, Tibetan Plateau, and other alpine regions) and habitats (snow, ice, and cryoconite), multiple testing was performed

based on the Dunn's post-hoc test using FSA package in the R environment (Ogle et al., 2022). The community structure variations were visualized using an NMDS ordination plot based on the Hellinger-transformed Bray-Curtis distance matrix. Permutational analysis of variance (PERMANOVA) was used to test the significance of community differences in samples by region and habitat (Anderson, 2017) using the "vegan" package in R with 999 permutations. PERMDISP analysis was performed using the betadisper command in the vegan package. Core phylotypes were defined as occurring in more than 55%

of the samples with average relative abundance > 0.1% in each habitat-region pair (Delgado-Baquerizo et al., 2018). If a phylotype was identified as a core phylotype for all habitats of a region, then it was designated as the core phylotype for the region. This classification was modified from, so that the dominant phylotype designation is less affected by the unbalanced samples for each habit-region pair.2.3 Metagenome data processing

Metagenome data processing has been described previously (Liu et al., 2022). Briefly, it includes raw data quality filtering,

assembly, open reading frames prediction, and genome binning. Gene open reading frames (ORFs) for the metagenomic assemblies were predicted using Prodigal (Hyatt et al., 2010), and dereplicated by clustering at 80% aligned region with 95% nucleotide identity using MMseqs2 (Steinegger and Söding, 2017) with parameters: easy-linclust -e 0.001, --min-seq-id 0.95, -c 0.80.

Metagenomic assemblies were binned using MetaBAT 2 (v2.12.1) (Kang et al., 2019), MaxBin 2 (v2.2.7) (Wu et al., 2016),

and VAMB (v2.0.1) (Nissen et al., 2021) separately. The resulting bins (or MAGs) were then refined using RefineM (v0.0.20) (Parks et al., 2017) by removing contigs with divergent GC content, coverage, or tetranucleotide signatures. Then only MAGs meeting the medium and higher quality of MIMAG (Bowers et al., 2017a) were retained (completeness > 50%, contamination <10%). The obtained MAGs were combined with the isolate genomes, and these genomes were dereplicated using the

thresholds of 10% aligned fraction and a genome-wide average nucleotide identity (ANI) threshold of 95%. They were then
taxonomically annotated using the Genome Taxonomy Database Toolkit (GTDB-Tk, v2.4) (Chaumeil et al., 2019) against the
GTDB release R220.

### 2.4 Gene function annotation

The functions of the dereplicated genes were also annotated using eggNOG-mapper (Huerta-Cepas et al., 2017) and the
eggNOG Orthologous Groups (OGs) database (v5.0) (Huerta-Cepas et al., 2019). This includes the KEGG functional orthologs
(Kanehisa et al., 2017), the carbohydrate-active enzymes database (CAZy) (Levasseur et al., 2013), and the COG categories
(Tatusov et al., 2003). Antibiotic resistance genes (ARGs) were annotated against the Comprehensive Antibiotic Resistance
Database (CARD) (Jia et al., 2017) and Resistance Gene Identifier (RGI v3.1.4) (Alcock et al., 2020) with the loose model (-
-include_loose). Virulence factors were annotated by aligning gene sequences against the Virulence Factors Database (VFDB
2019) (Liu et al., 2019) with DIAMOND blastp (Buchfink et al., 2021) (e-value threshold of $1e^{-5}$).

## 3 Results

### 3.1 Amplicon-based dataset

A total of 2,039 glacier-related samples were retained after quality filtering, comprising 1077 from cryoconite sediment, 601
from snow or ice, 216 from glacial melting water (including cryoconite hole meltwater and supraglacial melting water), 79
from ice core, 34 from snow during algal blooming (Algal material), and 32 from subglacial basal ice (**Table 1**). Spatially, 29%
of all samples (n = 574) were from the Antarctic, 28% (n = 574) from Arctic glaciers, 16% (n=335) from the Tibetan Plateau
(n = 335), and 27% from other alpine regions (n = 545, the Alps, Keniya, Japan, and Montana Glacier National Park).
The retained datasets are originated from 66 bioprojects, eight of which missed sequencing platform information and two of
which missed primer information (**Table S2**). Most of these bioprojects do not have environmental metadata, therefore are
not included in the dataset. A variety of primers were used by these projects, amplifying the hypervariable regions V3V4,
V4, and V4V5 regions. These data were harmonized by retaining only the V4 region (sequence trimming). Surprisingly, four
bioprojects that used primers 783F and 1046R (V5V6 region) were also retained. We speculate that incorrect primers may
have been provided in the NCBI.

Table 1 Number of samples by habitat and region.

|  | Snow | Ice | Cryoconite | Supraglacial meltwater | Ice core | Algal material | Basal ice |
|---|---|---|---|---|---|---|---|
| Antarctic | 119 | 58 | 335 | 59 | 0 | 1 | 13 |
| Arctic | 107 | 74 | 286 | 59 | 6 | 23 | 19 |
| Tibetan Plateau | 67 | 8 | 172 | 30 | 58 | 0 | 0 |
| Other non-polar glaciers | 89 | 79 | 284 | 68 | 15 | 10 | 0 |

### 3.1.1 Prokaryotic diversity

The Good's coverage index provides estimation for the number of singletons in a sample, reflects the coverage of the sequencing. The values of the index were 0.98±0.02 and 0.96±0.02 for the datasets without and with subsampling, respectively (Table S7). This indicates that majority of the OTUs were identified. The amplicon sequencing dataset comprised 64,510 phylotypes. Due to the large variation in sequencing depth among samples, here we only presented patterns that are consistently observed in the original and subsampled OTU tables (**Tables 2**, **3**, and **S7**). Across all habitats, the number of OTU observed (prokaryotic richness) was significantly lower in other non-polar glaciers than those in the three poles (Antarctic, Arctic, and Tibetan Plateau). This pattern is also significantly observed for the alpha diversity indices including the Chao1, ACE, and Gini Simpson, but not in the Shannon diversity. Specifically, the prokaryotic Shannon diversity of the Arctic glaciers was not significantly different from non-polar glaciers. Across habitats (**Table 3**), supraglacial meltwater had the highest prokaryotic diversity compared with other habitats (except the ice core). These alpha diversity indices typically follow orders of supraglacial meltwater > snow > ice core > ice > cryoconite > algae-influenced snow > basal ice. This may reflect the strength of environmental filtering among the habitats.

We further compared the alpha diversity indices of the same habitats across different regions (**Table S8**). Due to the bias in the number of samples across habitats, here we only compared the cryoconite, snow, ice, and supraglacial water. For cryoconites, their prokaryotic diversity (Shannon diversity and Gini Simpson indices) was the highest in the Antarctic, which was followed by the Arctic, Tibetan Plateau, and other non-polar glaciers. For snow, its prokaryotic diversity (Richness, Chao1, and ACE indices) was the highest in Tibetan glaciers, followed by Antarctic, Arctic, and other non-polar glaciers. For ice, the diversity indices were not significantly different across all regions, with only significantly higher values of diversity being observed in the Antarctic than those in the Arctic and Tibetan Plateau. For supraglacial meltwater, other non-polar glaciers demonstrated the highest diversity (Shannon diversity and Gini Simpson diversity indices, all $P < 0.05$), while the Antarctic had the lowest values. In summary, the biogeographic pattern for prokaryotic diversity is the same habitat differed among regions.

We further assessed the influence of the region amplified on the validity of the results. For each region-habitat pair, the alpha diversity indices were significantly different by these factors to a certain extent. However, those from Arctic basal ice, non-polar glacier ice, and Tibetan Plateau supraglacial meltwater were less affected (**Table S9**). Nevertheless, the influence may be explained by the different sampling locations, which have distinct microbial composition. We further tested the validity of

the diversity comparison results using the data that were generated using the same primer set (**Table S10**). The influence of primer selection on prokaryotic diversity was inconsistent. For instance, primers targeting the V4 region resulted in a higher richness in supraglacial ice than primers targeting the V3V4 region in other alpine glaciers. In contrast, the primers targeting the V3V4 region had a higher in the Arctic. Such inconsistency in microbial community assessment by different primers and platforms has been reported previously (Fredriksson et al., 2013; Tremblay et al., 2015). Thus, the homogenization method may not fully overcome the bias caused by primer selection. Nevertheless, we provided necessary information on the primer and regions amplified, user of the dataset may choose to use the entire dataset or only those using the same primer for analysis.

Table 2 Diversity metrices of the bacterial community in glacier microbiomes of global glaciers.

| Region | Antarctic | Arctic | Non-polar glacier | Tibetan Plateau |
|---|---|---|---|---|
| **Original dataset** | | | | |
| Species observed | $1025.1\pm722.5^b$ | $981.4\pm784.4^b$ | $868\pm676.3^c$ | $1185\pm825.9^a$ |
| Chao1 | $1400\pm912.5^b$ | $1492.4\pm1308.7^b$ | $1223.2\pm941.8^c$ | $1768.2\pm1279.3^a$ |
| ACE | $1422.5\pm948.1^b$ | $1526.9\pm1340.8^{ab}$ | $1232.3\pm959.5^c$ | $1767.5\pm1295.4^a$ |
| Shannon diversity | $4.3\pm1.03^a$ | $3.97\pm1.05^b$ | $4.17\pm1.19^b$ | $4.2\pm1.3^a$ |
| **Subsampled dataset** | | | | |
| Region | Antarctic | Arctic | Non-polar glacier | Tibetan Plateau |
| Species observed | $475.7\pm232^a$ | $467.9\pm275.9^a$ | $468.1\pm384.2^b$ | $507.1\pm317.3^a$ |
| Chao1 | $809.6\pm458.3^a$ | $847.9\pm575.3^a$ | $777.5\pm679.4^b$ | $884.4\pm622^a$ |
| ACE | $825.7\pm482.6^b$ | $881.6\pm619.9^{ab}$ | $796.6\pm715.3^c$ | $903.4\pm651.8^a$ |
| Shannon diversity | $4.23\pm1^a$ | $3.9\pm1.02^b$ | $4.1\pm1.16^b$ | $4.11\pm1.25^a$ |

Statistical test is based on Kruskal-Wallis one-way ANOVA, multiple testing is performed based on the Dunn's post-hoc test. Different letters indicate significant differences at $P = 0.05$.

Table 3 Diversity metrices of the bacterial community in glacier microbiomes of different habitats.

| Habitats | Algal material | Basal ice | Cryoconite | Supraglacial Ice | Ice core | Snow | Supraglacial meltwater |
|---|---|---|---|---|---|---|---|
| **Original dataset** | | | | | | | |
| Species observed | 596.3±451.7$^{ab}$ | 489.6±306.3$^{a}$ | 904.4±532.6$^{c}$ | 929.2±695.9$^{bc}$ | 999.9±438.3$^{cd}$ | 1178.8±1121.6$^{c}$ | 1344.3±910.3$^{d}$ |
| Chao1 | 873.5±648.2$^{ab}$ | 646.8±379.1$^{a}$ | 1316.9±786.1$^{c}$ | 1285.8±962.3$^{bc}$ | 1452.8±618.7$^{cd}$ | 1726.4±1736.5$^{bc}$ | 1898.4±1307.7$^{d}$ |
| ACE | 905.1±682.1$^{ab}$ | 641.8±388.3$^{a}$ | 1335.5±806.4$^{c}$ | 1296±979$^{bc}$ | 1430.8±599$^{cd}$ | 1760.1±1777.7$^{bc}$ | 1914.2±1339.1$^{d}$ |
| Shannon diversity | 3.18±1.38$^{a}$ | 3.4±0.99$^{a}$ | 4.21±0.84$^{b}$ | 3.67±1.09$^{a}$ | 4.6±0.8$^{c}$ | 3.8±1.5$^{a}$ | 5±1.2$^{c}$ |
| **Subsampled dataset** | | | | | | | |
| Species observed | 317.5±293$^{ab}$ | 239.2±129.5$^{a}$ | 464.8±223$^{c}$ | 356.1±205.1$^{ab}$ | 521.7±207.9$^{cd}$ | 471.9±386.5$^{bc}$ | 710.1±458.2$^{d}$ |
| Chao1 | 568.6±544.6$^{abc}$ | 357.4±201.4$^{a}$ | 798.9±424.1$^{d}$ | 613.7±405.5$^{b}$ | 799.5±325.4$^{de}$ | 883.2±807.7$^{cd}$ | 1176.4±836.6$^{e}$ |
| ACE | 588.9±581.2$^{abc}$ | 353.4±207.7$^{a}$ | 818.5±442.6$^{d}$ | 626.1±419.2$^{b}$ | 795.6±339.9$^{de}$ | 921.9±868.7$^{cd}$ | 1207.6±888.5$^{e}$ |
| Shannon diversity | 3.14±1.35$^{a}$ | 3.37±0.99$^{a}$ | 4.14±0.82$^{b}$ | 3.6±1.06$^{a}$ | 4.5±0.8$^{c}$ | 3.7±1.4$^{a}$ | 4.9±1.1$^{c}$ |

Statistical test is based on Kruskal-Wallis one-way ANOVA, multiple testing is performed based on the Dunn's post-hoc test. Different letters indicate significant differences at $P = 0.05$.

### 3.1.2 Prokaryotic taxonomy composition in the glaciers of the Three Poles

We identified 53 bacterial and archaeal phyla across the dataset. The glacier microbiomes had similar taxonomy composition, dominated by Proteobacteria (averaged 37.8%), Cyanobacteria (22.2%), Bacteroidetes (20.3%), and Actinobacteria (9.2%) (**Fig. 2** and **Tables S11**). At the class level (**Tables S12**), the microbiome was dominated by Gammaproteobacteria (22.7%, Proteobacteria), Oxyphotobacteria (22.1%, Cyanobacteria), Bacteroidia (22.2%, Bacteroidetes), Alphaproteobacteria (13%, Proteobacteria), and Actinobacteria (class) (8.1%, Actinobacteria).

### 3.1.3 Bacteria community structure in the glaciers of the Three Poles

NMDS ordination plot revealed distinct prokaryotic communities among habitats (**Fig. 3a**). PERMANOVA analysis showed significantly different prokaryotic community structures among Antarctic, Arctic, Tibetan Plateau, and other non-polar glaciers ($P < 0.001$). The influence of location exhibited a higher $R^2$-value (0.10535) than the influence of habitat ($R^2 = 0.06299$), suggesting that spatial location may have a greater influence in shaping glacier microbiomes. PERMDISP analysis showed that snow samples exhibited a significantly (at the threshold of $P = 0.05$) higher dispersal from centroid (67.4%) compared to cryoconite (65.1%). Comparatively, those of ice core and algae were lower (57.9% and 58.9%, respectively), possibly due to the low sample numbers.

We identified the top three most abundant phylotypes for each habitat-region pair (**Fig. 3b** and **Table S13**). These abundant phylotypes were mainly affiliated with Proteobacteria, Cyanobacteria, and Bacteroidetes. The distribution of these abundant phylotypes clustered predominantly by geographical location, reflecting strong spatial effects. We then attempted to identify ubiquitous phylotypes for each region-habitat pair, defined as those present in more than 55% of samples with a relative abundance greater than 0.1%. However, we found no phylotypes that were common across any regions or habitats (**Table S14**).

The number of ubiquitous phylotypes varied, ranging from nine in Arctic ice to 129 in Tibetan-supraglacial meltwater. Notably, we did not identify any ubiquitous phylotypes in Arctic-snow or other non-polar-snow. Most of the ubiquitous phylotypes were associated with Gammaproteobacteria (29% of the total), followed by Bacteroidetes (19%), Alphaproteobacteria (16%), Cyanobacteria (13%), and Actinobacteria (11%). This distribution may indicate their ability to disperse and adapt to different environmental conditions.

**3.2 Metagenome- and genome-dataset**

We acquired 226 glacier metagenome data (**Table S3**) and 999 bacterial genomes from glacial environments (**Table S4**). After quality filtering and assembly, 63,294,073 unique Open Reading Frames (ORFs) were obtained. Of these dereplicated ORFs, 47.8% (29,947,128) were functionally annotated using eggNOG.

**3.2.1 Overall features glacier metagenome-assembled genomes**

After binning, the dataset generated 3,502 metagenome-assembled genomes of medium quality (Genome completion $\geq$ 50%, contamination < 10%) and higher (Bowers et al., 2017b). After combining the genomes of cultivated glacier bacteria (999), this expanded the total genome number to 4,501 from the previously published 3,322 (Liu et al., 2022) (**Table S15**), a 35.5% increase. The median genome size was 3.46 Mb ranging from 0.42 Mb to 10.49 Mb; the GC% was 60%, ranging from 30% to 76%. The MAGs were taxonomically affiliated with 33 phyla, 79 classes, 154 orders, 271 families, and 549 genera (**Fig. 4a**).

Additionally, 3470 (77.1% of all MAGs) were unable to be classified at the species level (**Fig. 4b**), reflecting substantial genomic novelty in the glacier microbiome. These genomes were dereplicated into 1400 genomic OTUs (gOTUs), which are typically considered species.

**3.2.2 Key functional genes**

**Carbohydrate-active enzymes**: The dataset contains 1,082,125 genes encoding carbohydrate-active enzymes (CAZymes, **Fig.**
**5a**), i.e., those enzymes involved in the metabolism of glycoconjugates, oligosaccharides, and polysaccharides (Zerillo et al., 2013). Genes associated with carbohydrate hydrolysis (GH) and biosynthesis (GT) were the most abundant, accounting for 45.2% and 44.4%, respectively. In contrast, those genes associated with non-hydrolytic cleavage of glycosidic bonds (PL), hydrolysis of carbohydrate esters (CEW), and assisting in degrading biomass substrates (AA) were relatively scarce, accounting for 0.8%, 3.1%, and 0.2% of the predicted CAZY, respectively. This indicates that the glacier microbiome is
competent in a diverse range of carbon transformation processes, mediating the delivery of carbon to downstream ecosystems.

We further examined the distribution of CAZyme genes across different glaciers. At the CAZY Family level, GT2, GT4, GH13, and GT51 are the most diverse across all habitats (**Fig. 5b**). PERMANOVA tests revealed significant composition differences by both habitat (pseudo-F = 10.014, $P < 0.001$) and regions (pseudo-F = 5.038, $P = 0.002$), with habitat exhibiting a greater influence on CAZY composition. Specifically, the PCA plot revealed a greater number of genes classified as GH5 and GH9 in cryoconites (**Fig. 5c**). Furthermore, 30 GHs and 20 GTs exhibited significantly higher contribution in cryoconites than in ice or snow (**Fig. 5d**). Comparatively, 15 GHs and 27 GTs exhibited significantly higher contribution in ice or snow than in cryoconites. Thus, cryoconites exhibited higher capacity in the catabolism of organic carbon, whereas snow and ice are dominated by anabolism.

**Nitrogen cycling:** The dataset contained 138,421 unique genes associated with nitrogen cycling, most of which (99.3%) were associated with nitrate reduction and/or denitrification pathways (**Fig. 6a**). These genes included the *nirB* gene responsible for the nitrite reduction to ammonia in the assimilatory nitrate reduction pathway, the *narB* and *nirA* genes responsible for sequential nitrate reduction to ammonia in dissimilatory nitrate reduction pathways, and the *nirK* gene responsible for nitrite reduction to nitric oxide in denitrification pathway. This suggests that microbial-driven nitrate reduction is widespread in glacial habitats for both nitrogen assimilation and energy supply, highlighting their potential roles in $NO_x$ formation. In comparison, genes involved in nitrogen fixation (*nifH*) and nitrification (*hao*) were relatively rare, with only 678 (0.49% of the nitrogen cycling-related genes) and 240 (0.17%) unique genes identified, respectively. This suggests that microorganisms capable of these high-energy demand processes only account for a small fraction of the glacial microbiome, which is consistent with the low nitrogen fixation rates reported in glacier-related habitats (Telling et al., 2011).

The differences in gene distributions are apparent among different habitats. Specifically, the genes associated with nitrogen fixation and denitrification are mainly present in cryoconites. The former could be explained by the high abundance of Cyanobacteria (**Fig. 2**), while the latter could be attributed to its anaerobic conditions. Thus, the cryoconite could be a potential source of $N_2O$. Comparatively, *amoA* (in nitrification) is mainly identified from ice, while *norB* and *nosZ* (in denitrification) are also abundant. This reflects that ice could harbor greater functional diversity than snow, with the capacity for nitrogen transformation and influencing the nutrient discharged to downstream ecosystems.

**Methane cycling:** The dataset contained 154 methane cycling-related genes. Of these, 93 were the soluble form of methane oxidase (*mmoX*), accounting for 61% of the total methane-cycling genes identified (**Fig. 6b**). More *mmoX* genes were identified from cryoconite (34.4% of the total methane-cycling genes identified) than from ice (20.8%) or snow (5.2%). Conversely, genes associated with the particulate form of methane oxidation (*pmoA*, *pmoB*, and *pmoC*) were more frequently identified from ice (21.4%) than from cryoconite (5.8%) and snow (3.2%). The two forms of methane monooxygenase (sMMO and pMMO) have distinct enzymatic characteristics and are expressed in different growth conditions. Specifically, sMMO is expressed under low copper conditions ($\leq 0.9$ nmol of Cu/mg of cell protein), while pMMO is expressed under relatively high copper/biomass ratios (Zhang et al., 2017). Furthermore, sMMO has a broader substrate range, can oxidize methane, short chain alkane, alkene, and aromatic compounds, while pMMO can only oxidize alkanes of less than five carbons (Trotsenko and Murrell, 2008). Despite being not empirically measured, the cryoconite is expected to contain a higher concentration of

copper than the ice, as the former is precipitated dust. Thus, the higher diversity of sMMO in the cryoconite could be associated with its ability to oxidize diverse alkanes, which may be present in soils. In comparison, high-affinity pMMO could oxidize methane at atmospheric levels, which may support microbial communities in the oligotrophic ice habitat. Only six unique methanogenesis-related genes (*mcrA*) were identified, almost exclusively in cryoconite metagenomes. This is consistent with the cryoconite as a methane source in the literature (Zhang et al., 2021).

**Antimicrobial resistance genes**: Using thresholds of 80% identity and 80% sequence coverage, we identified 1166 ORFs that exhibited high sequence similarity to 224 antibiotic resistance genes (ARG). Of these identified ARGs, *MexF*, beta-lactamase, and *mexK* were the most abundant, accounting for 8.1%, 4.2%, and 3.7% of the ARGs identified, respectively (**Table S16**). The predominant antibiotic resistance mechanisms were antibiotic efflux and antibiotic inactivation, accounting for 44% and 41% of the total ARGs identified, respectively. These ARGs were predicted to confer resistance against 30 different antibiotics, with penam, tetracycline, and macrolide being the most commonly encountered resistant targets. Additionally, 54% of the identified ARGs provided multiple drug resistance, with the *OprM*, *CpxR*, and *tolC* genes conferring resistance to 16, 15, and 15 types of antibiotics, respectively. ARGs were identified in 566 genomes (13% of total genomes obtained). This low proportion of ARG-bearing genomes suggests that the glacier habitats are only weakly affected by antibiotic contamination. Of the genomes containing ARGs, 48.6% and 34.1% were affiliated with Proteobacteria and Firmicutes, respectively (**Fig. 6c**). However, the resistance mechanisms exhibited by these two bacterial phyla were markedly distinct, with antibiotic efflux (*MexF*) and antibiotic target alternation (*vanZf*)/inactivation (*FosB*) being the most common mechanisms for Proteobacteria and Firmicutes, respectively. Most of the genomes (n=224) carried only a single ARG, while seven genomes possessed more than ten ARG genes, with *Pseudomonas aeruginosa* genomes hosting up to 48 ARGs.

**Virulence factors**: Using thresholds of 80% identity and 80% coverage, the dataset contains 66,822 virulence factor genes, accounting for 0.11% of the total ORFs identified (**Table S17**). Virulence factors were predominately associated with adherence, motility, and immune modulation functions, while those associated with toxin production accounted for only 0.48% (**Fig. 6d**). We did not detect any toxin genes from the genomes obtained using the same thresholds, with only those associated regulation, effector delivery systems, and metabolic factors being identified from Proteobacteria, Actinobacteria, and Deinococcota genomes. Nevertheless, 878 potential toxin genes were identified from the genomes if the criteria were loosened, with sequence identified ranging from 20.1% to 67.8%, these genes may represent novel toxins without references in the dataset, or non-toxin genes homologues to known toxin genes. These candidate toxin genes were most abundantly identified in Gammaproteobacteria, followed by Bacteroides (15.9%) and Alphaproteobacteria (10.0%).

## 4 Data availability

The data introduced here is the first step in archiving global glacier microbial data. For this purpose, the data is deposited into the Global Glacier Genome and Gene Database (4GDB, https://nmdc.cn/4gdb/) and the National Tibetan Plateau Data Center (https://doi.org/10.11888/Cryos.tpdc.300830, Liu, et al., 2023), which provides a comprehensive solution for glacier microbial

studies, featuring amplicon sequencing phylotype table, representative sequences, taxonomic annotations, metagenomic raw sequences, assembled contigs, annotated gene sequences, sequences of metagenome-assembled genomes, and the growth characteristics of cultivated microorganisms, into a user-friendly website. The website is constructed Under the OSI-approved

CC BY 4.0 Open Source license (https://creativecommons.org/licenses/by/4.0/), all data can be accessed and reused freely, without any restrictions, for both academic and commercial purposes.

The 4GDB website is mainly structured into three sections, comprising amplicon sequencing, metagenome/genome sequences, and function prediction. The user-friendly web interface allows data filtering based on sample type, sample location, habitat type, gene type, and taxonomy, enabling seamless download of the filtered results. In conclusion, 4GDB

(https://nmdc.cn/4gdb/downloadtemp) provides an open-access genome- and gene-orientated resource platform that is regularly updated to include newly published and in-house generated sequence data.

## Author contributions

YL conceptualized the paper, SH, YL, TY, ZZ, YC, KL, PL, JL, and MJ analysed the data, TY, SG, QS, GF, LW, and JM

developed the website, MJ and YL prepared the manuscript with contributions from all authors.

## Competing interests

The contact author has declared that none of the authors has any competing interests.

## Acknowledgements

This study was supported by the National Natural Science Foundation of China 42421001 (YL) and 42425607 (MJ); and the

Department of Science and Technology, Gansu, China 25RCKA021 (MJ). Tables S5 and S6 are deposited in Figshare.com (DOI: 10.6084/m9.figshare.28423781).

**Figures and figure legends**

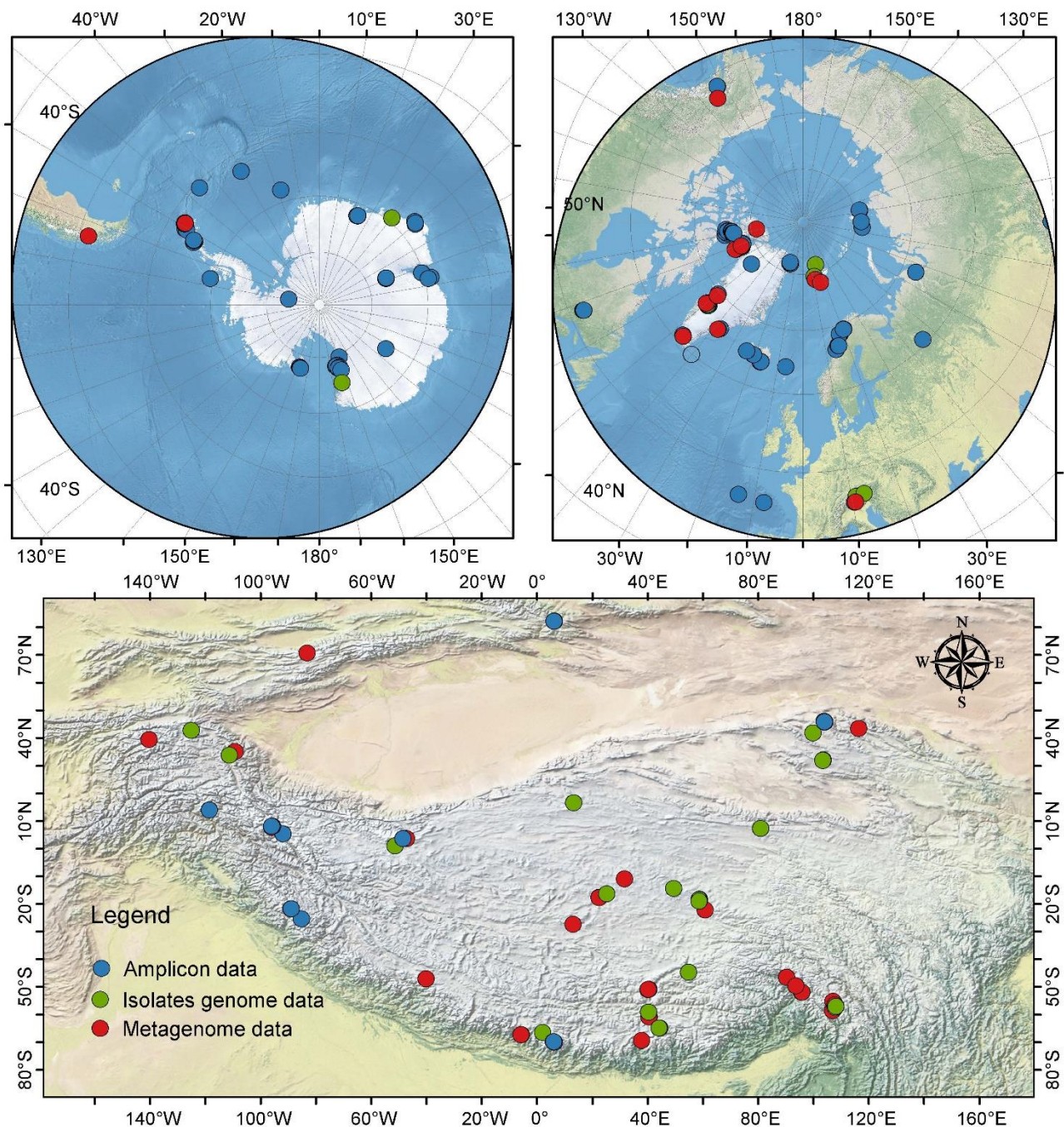

**Fig. 1 The location of glacier samples across the Antarctica, Arctic, and Tibetan glaciers**

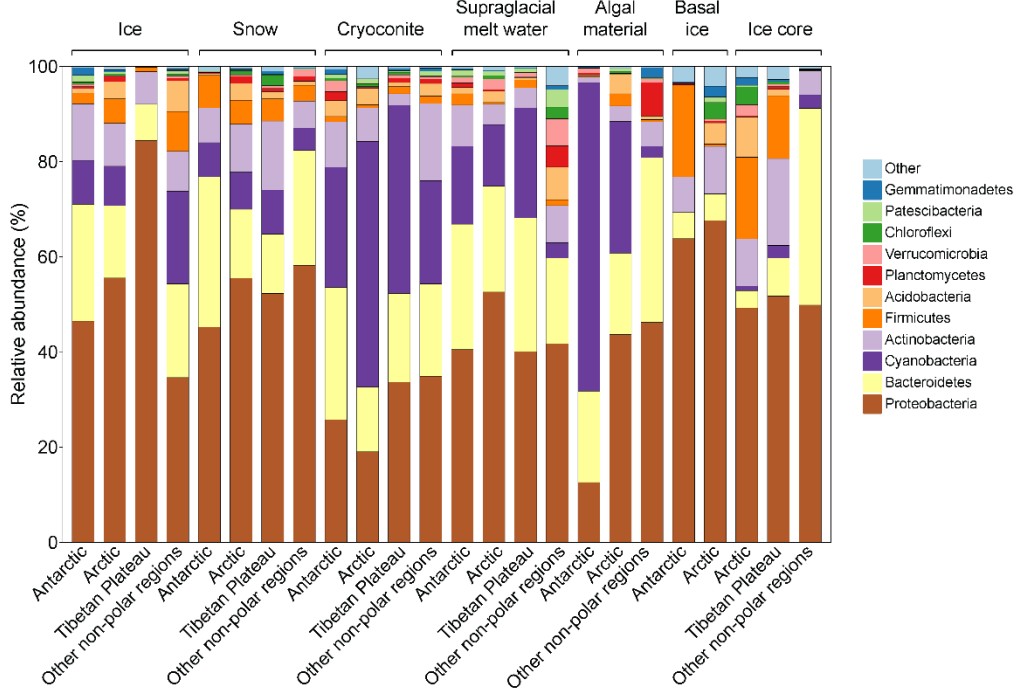

Fig. 2 Taxonomy composition of supraglacial ecosystems in glaciers of the Antarctic, Arctic, Tibetan Plateau, and other alpine regions.

Taxonomy is inferred based on amplicon sequencing results.

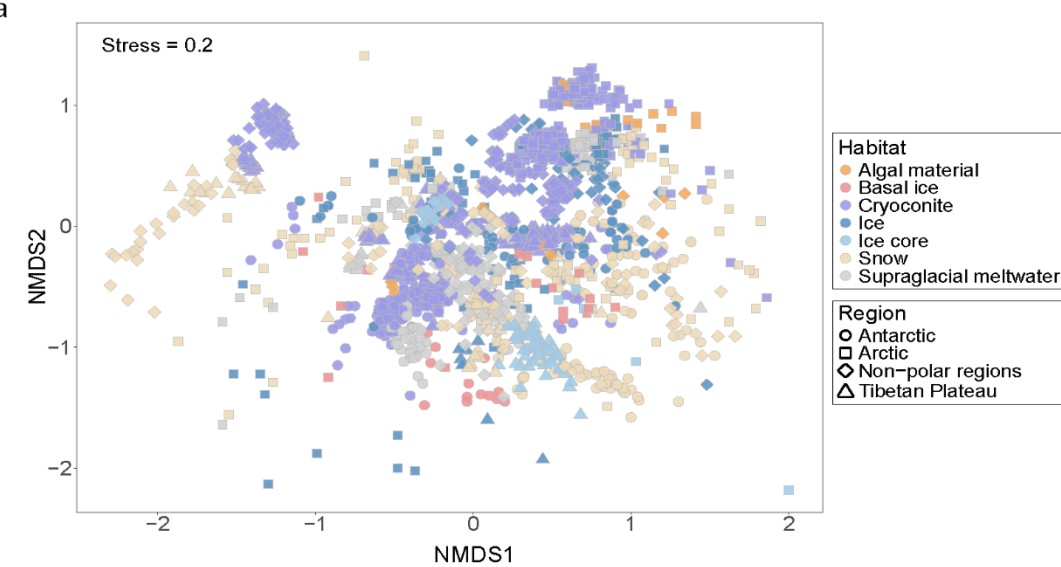

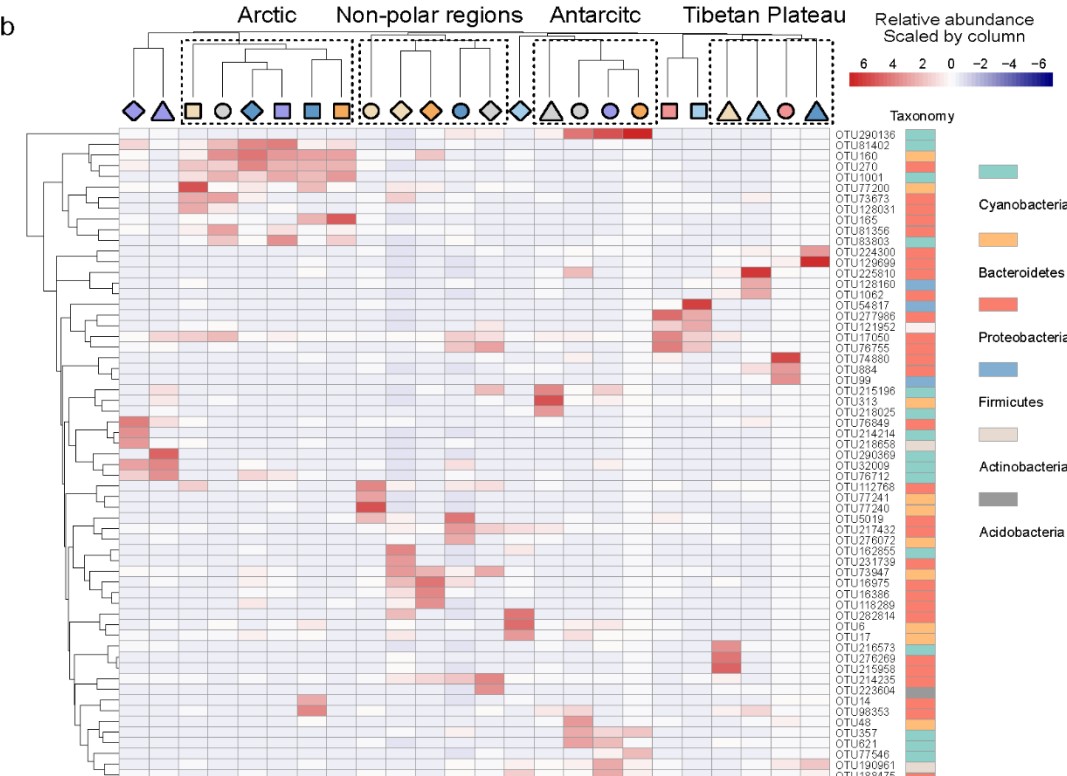

**Fig. 3 The community structure of glacier microbiomes across the Antarctica, Arctic, and Tibetan Plateau.**

(a): Microbial community structure differences visualized using the non-metric multidimensional scaling ordination plot; (b):

The heatmap highlights the distribution pattern of dominant phylotypes for each habitat-region pair.

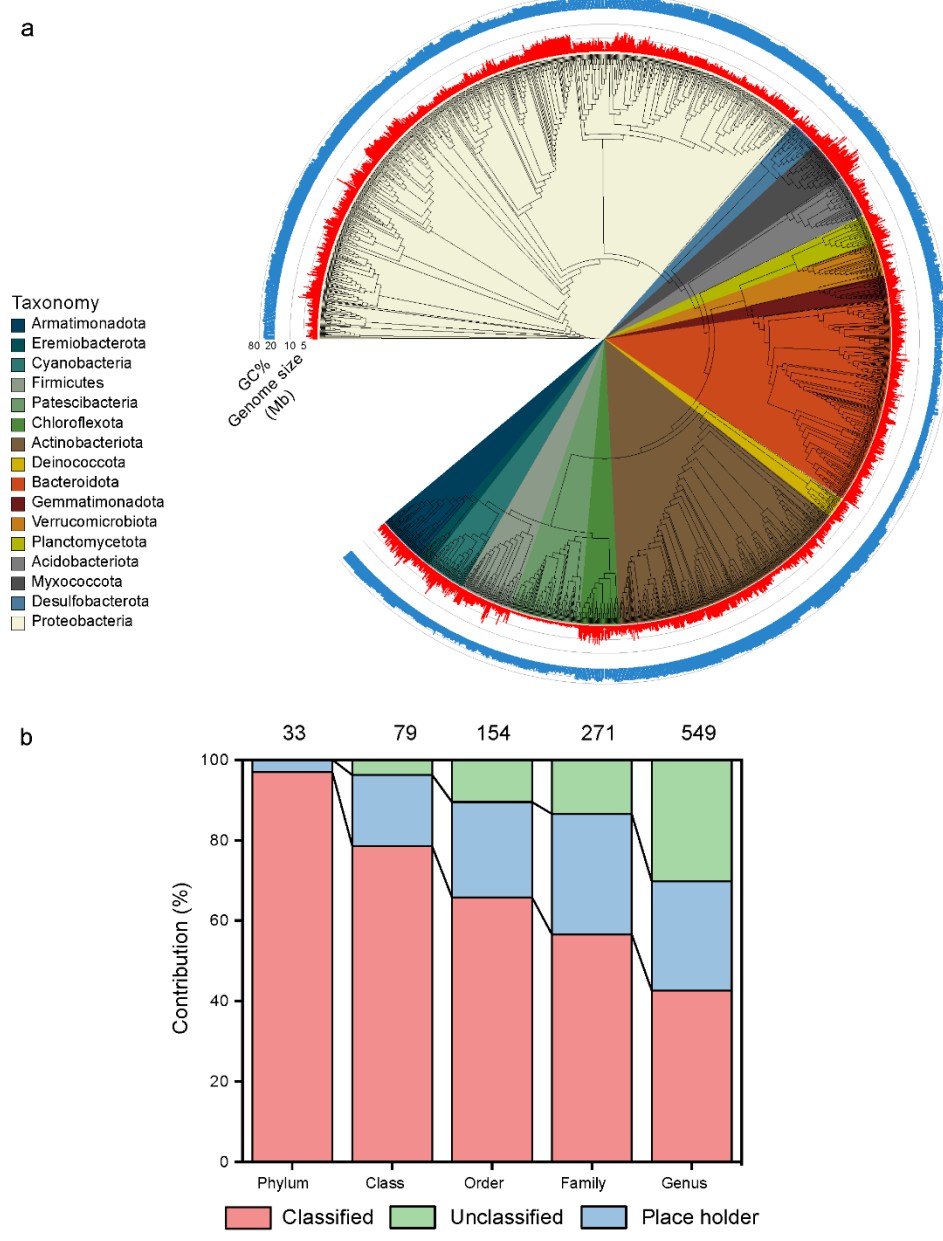

Fig. 4 Taxonomy classified of the obtained prokaryotic genomes

A: The phylogenetic tree contains the representative bacterial genomic OTUs with genome size and GC% being shown; b: The taxonomic classified of the obtained genomes. Classified refers to valid taxonomic classification, place holders are genomes that have been deposited in the GTDB R220 database, while unclassified are genomes that have not been deposited in the GTDB database.

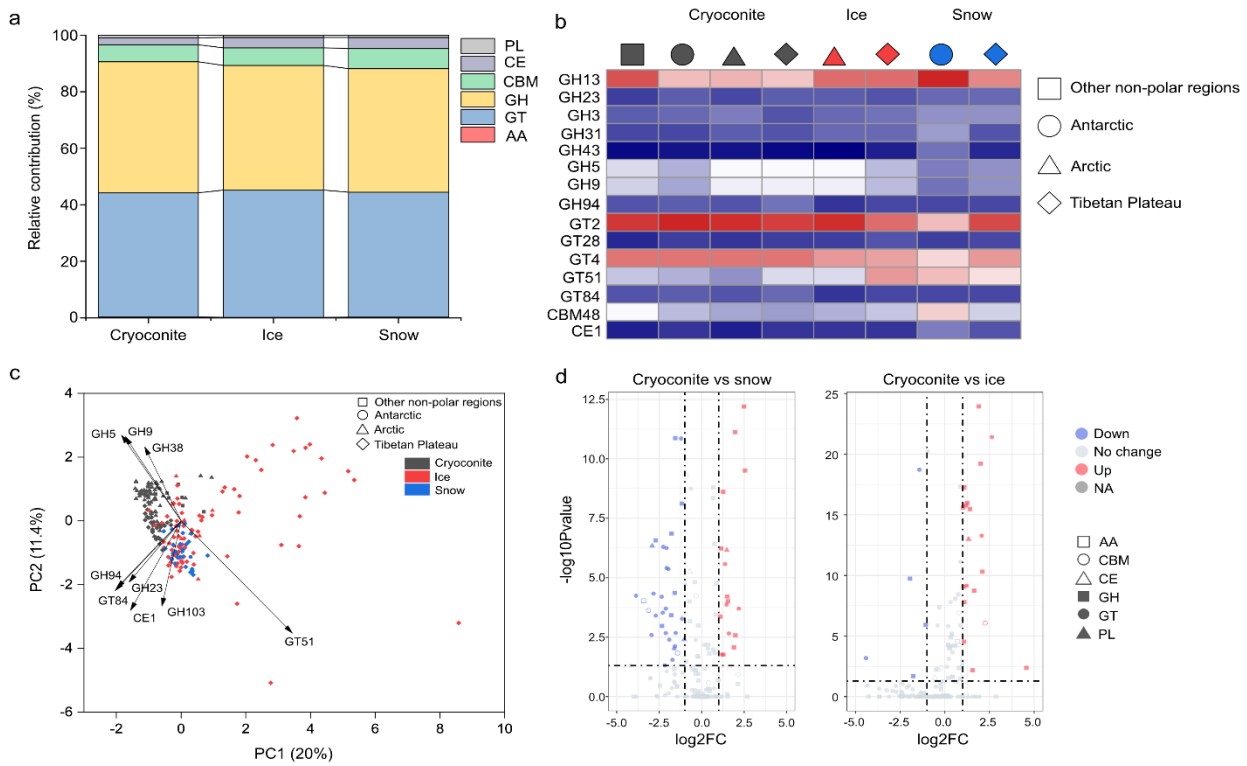

**Fig. 5 The distribution of genes associated with carbohydrate-active enzymes (CAZymes) in glaciers.**

a: The relative contributions of genes associated with different CAZymes in each habitat (GH: Glycoside hydrolases; GT: glycosyl transferases; PL: Polysaccharide lyases; CE: Carbohydrate esterases; AA: Auxiliary activities); b: The most diverse CAZymers in each habitat-region combination. c: PCA plots shows the distribution of CAZymes across habitats and regions; d: the selective enrichment of CAZymes between cryoconite and snow and between cryoconite and ice.

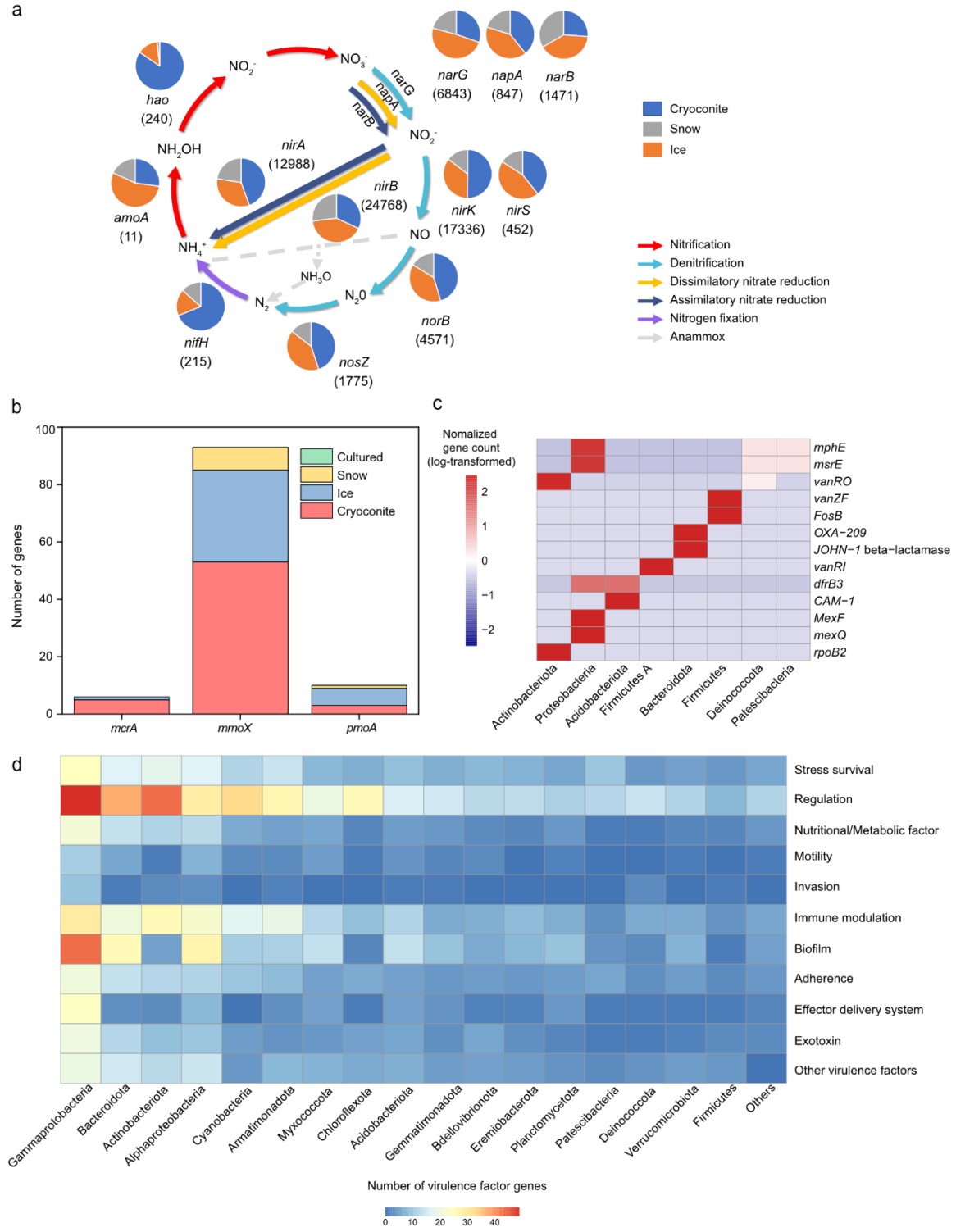


**Fig. 6 Features of the functional genes across the glacier metagenomes from the Antarctica, Arctic, and Tibetan Plateau.**

a: nitrogen-cycling (The numbers indicate the number of genes identified, gene of the anaerobic ammonium oxidation pathway are not identified in the glacier metagenomes); b: methane cycling (*mcrA* gene is responsible for methanogenesis; *mmoX* gene is the soluble form methane oxidation gene, while *pmoA* is the particulate form methane oxidation gene, both of which are associated with methane oxidation); c: genes associated with antibiotic resistance; and d: the number of genes associated with virulence factors, the numbers have been square root-transformed.

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
