# Peer review of "A database of glacier prokaryotic genomes and genes for the Three Poles"

_Earth System Science Data, 2023_

## Community Comment (CC1)

[Figure]

(a)

2D stress = 0.17

**Regions**
- Antarctic
- Arctic
- Tibetan Plateau

**Habitats**
- □ Cryoconite
- ○ Cryoconite water
- ▽ Ice
- △ Snow

NMDS2

NMDS1

(b)

Log transformed
relative abundance

OTU1037 (Cyanobacteria;Cyanobacteriia;Cyanobacteriales)
OTU433 (Cyanobacteria;Cyanobacteriia;Cyanobacteriia_unclassified)
OTU10331 (Proteobacteria;Gammaproteobacteria;Burkholderiales)
OTU1855 (Bacteroidota;Bacteroidia;Sphingobacteriales)
OTU43616 (Proteobacteria;Gammaproteobacteria;Burkholderiales)
OTU2564 (Bacteroidota;Bacteroidia;Flavobacteriales)
OTU47186 (Proteobacteria;Gammaproteobacteria;Pseudomonadales)
OTU677 (Proteobacteria;Alphaproteobacteria;Sphingomonadales)
OTU1424 (Proteobacteria;Alphaproteobacteria;Sphingomonadales)
OTU2385 (Bacteroidota;Bacteroidia;Bacteroidia_unclassified)
OTU1205 (Proteobacteria;Gammaproteobacteria;Burkholderiales)
OTU13604 (Proteobacteria;Gammaproteobacteria;Pseudomonadales)
OTU37239 (Cyanobacteria;Cyanobacteriia;Cyanobacteriia_unclassified)
OTU1600 (Cyanobacteria;Cyanobacteriia;Cyanobacteriales)
OTU1019 (Cyanobacteria;Cyanobacteriia;Cyanobacteriales)
OTU2432 (Actinobacteriota;Actinobacteria;Micrococcales)
OTU14938 (Cyanobacteria;Cyanobacteriia;Cyanobacteriales)

Cryoconite sediment (Arctic)
Cryoconite sediment (Antarctic)
Cryoconite water (Antarctic)
Snow (Arctic)
Snow (Tibetan Plateau)
Ice (Tibetan Plateau)
Cryoconite sediment (Tibetan Plateau)
Cryoconite water (Tibetan Plateau)

---

## Author Response (AR1)

**Reviewer 1**

We are extremely thankful for these constructive comments, we have carefully addressed all comments and suggestions, please see detailed revisions below.

**Comment 1**

Line 35, add related references on "...... a pool of carbon and nitrogen". How about the carbon and nitrogen storage in the glaciers?

*Response*

The organic carbon stored in the global glacier was estimated on an order of 6 Pg (Hood et al., 2015), but the estimate for nitrogen stored in the global glacier is not yet available. We have amended the sentence for clarification

*Amended manuscript*

It has been estimated that six Pg of carbon are stored in global glaciers. These carbon may be released into downstream ecosystems with glacier runoff (Hood et al., 2015), influencing key elemental cycling in downstream ecosystems.

**Comment 2**

Line 35, "The six Pg of carbon...with glacier runoff." Do you really mean that every year six PgC can be released due to glacial runoff? This sentence is not so clear?

*Response*

This is an estimated number of the total amount of carbon stored in global glaciers. Certainly, not all glaciers will melt and release the carbon stored within. We have rewritten the sentence to clarify this.

*Amended manuscript*

It has been estimated that six Pg of carbon are stored in global glaciers. These carbon may be released into downstream ecosystems with glacier runoff (Hood et al., 2015), influencing key elemental cycling in downstream ecosystems.

**Comment 3**

Line 44-54, how about the "englacial and subglacial ecosystems" as you mentioned in the first sentence of this paragraph?

*Response*

We have added data from the ice core and subglacial ice. Therefore, englacial and subglacial ecosystems are covered in the dataset. We have also added additional sentences to elaborate on the activity differences between microorganisms in supraglacial ecosystem and englacial ecosystems.

*Amended manuscript*

Compared with other glacier-related habitats, the microorganisms in supraglacial ecosystems are the most active, due to their exposure to external environment and ambient temperature.

**Comment 4**

Line 55, Is the glacier surface meltwater belonged to "supraglacial ecosystem"?

*Response*

Glacier surface melting water is a part of the supraglacial ecosystem.

**Comment 5**

Line 61, Add a "." at the end of this sentence.

*Response*

We have added the missing "." at the end of the sentence.

**Comment 6**

Figure 2(a), the legends for Ice (Inverted triangle) and Snow (Regular triangle) should be separated.

*Response*

We have increased the space between the figure legends so that these symbols are separated.

[Figure]

**Comment 7**

Figure 3, What is the captions for (a) and (b) respectively? Do you mean they have the same captions, and the meaning of a,b,c on the figure are also the same?

*Response*

The results on potential pathogens have been removed by the comments from reviewer #3.

**Reviewer 2**

The manuscript offers a dataset on microbial communities from glaciers in Antarctica, the Arctic, and Tibet, comprising 815 amplicon sequence data, 952 cultured bacterial genome data, and 208 metagenomic data. This dataset, covering diverse habitats like ice, snow, and cryoconite, is instrumental in understanding microbial diversity, taxonomy, community structure, and genetic function in glacial environments. However, there are critical issues concerning the dataset's completeness need to be addressed.

*Response*

We appreciate the reviewer's suggestions, and we have provided detailed additional information on the sediment sampling process, please see comments below.

**Comment1**

One major concern is the apparent lack of comprehensiveness in the metagenomic data presented in Table S2. Notably, the authors' previous study (Zhang et al., 2023, Microbiome, https://doi.org/10.1186/s40168-023-01621-y), which analyzed 88 metagenomes from 26 glacier cryoconites, is not included. This omission is surprising and detracts from the dataset's perceived completeness.

*Response:*

The 88 metagenomes in Zhang et al., (2023, Microbiome) are all from published papers, all these dataset have been included in the present work.

They include

Franzetti A,Tagliaferri I,Gandolfi I, et al. Light-dependent microbial metabolisms drive carbon fluxes on glacier surfaces[J]. ISME J,2016,10: 2984-2988.

Bellas C M,Schroeder D C,Edwards A, et al. Flexible genes establish widespread bacteriophage pan-genomes in cryoconite hole ecosystems[J]. Nat Commun,2020,11: 4403

Zhang B,Chen T,Guo J, et al. Microbial mercury methylation profile in terminus of a high-elevation glacier on the northern boundary of the Tibetan Plateau[J]. Sci Total Environ,2020,708: 135226

Hauptmann A L,Sicheritz-Pontén T,Cameron K A, et al. Contamination of the Arctic reflected in microbial metagenomes from the Greenland ice sheet[J]. Environmental Research Letters,2017,12: 074019

Liu YQ, Ji MK, Yu T, et al., A genome and gene catalog of glacier microbiomes. Nature Biotechnology, 2022. 40(9): 1341.

Murakami T, Takeuchi N, Mori H,et al., Metagenomics reveals global-scale contrasts in nitrogen cycling and cyanobacterial light-harvesting mechanisms in glacier cryoconite. MICROBIOME, 2022. 10(1).

**Comment 2**

Furthermore, the absence of citations for other significant metagenomics studies (below) is a critical oversight, especially given the reliance of the current paper's results on metagenomic analysis. Incorporating a broader range of relevant literature is essential to bolster the study's credibility and thoroughness.

Varliero et al. (2021) Frontiers in Microbiology, doi.org/10.3389/fmicb.2021.627437
Melanie C. Hay et al., (2023) Microbial genomics, doi.org/10.1099/mgen.0.001131
Bellas et al. (2020) Nature Communications. doi.org/10.1038/s41467-020-18236-8
Edwards et al. (2013) Environ. Res. Lett. DOI 10.1088/1748-9326/8/3/035003

*Response*

We thank you for reminding us of the oversight of these important references.

- For Varliero et al. (2021) Frontiers in Microbiology, doi.org/10.3389/fmicb.2021.627437. We have included the six metagenomes from glacier habitats from this study (PRJEB41174, ERR4837082-ERR4837084, ERR4837105, ERR4837106, and ERR4837128).

- For Melanie C. Hay et al., (2023) Microbial genomics, doi.org/10.1099/mgen.0.001131. We have included six samples from this study is from the glacier habitats, we have included these in our dataset (PRJEB59067, ERR10878199-ERR10878204)

- For Bellas et al. (2020) Nature Communications. doi.org/10.1038/s41467-020-18236-8, we have included six metagenomes that are available from NCBI SRA archive in our dataset (SRR8842250, SRR8842249, SRR8842248, SRR12327455, SRR12327363, and SRR12350504). But for mgm4491734.3, it is deposited in MG-RAST and the server is no longer active. We could not retrieve this data anymore.

- For Edwards et al. (2013) Environ. Res. Lett. DOI 10.1088/1748-9326/8/3/035003. The data is deposited in MG-RAST, for the same reason, this is no longer available.

These additional metagenomes have been added to the current dataset.

**Comment 3**

Regarding the amplicon sequencing data in Table S1, the quantity appears insufficient for a study claiming to be a comprehensive database of DNA data from glaciers. To enhance the robustness of the dataset, it is advisable to include data from at least 10 or more amplicon sequencing studies focusing on glaciers. The current dataset's overrepresentation of data from the Tibetan Plateau might introduce a geographical bias in the study's findings.

*Response:*

We thank you for raising this concern. We performed an additional search in the NCBI Biosample database, using the keyword <Glacier OR ice OR snow OR cryoconite>, with sample type being DNA and instrument of Illumina, we retrieved 225,378 SRA entries initially. Then the results were carefully filtered manually to remove non glacier habitats (such as glacier forefield and ice cave), metagenome data, and primers that do not amplify the V4 region of the 16S rRNA gene (i.e., only those amply V3V4, V4, and V4V5 were retained) (Table S1).

We then only kept samples with more than 5000 reads for analysis. This ended with 2039 samples from 66 bioprojects.

Please see the Table 1 in the attached file.

This ended with similar number samples for both region and major habitats (snow, ice, cryoconite, and supraglacial meltwater).

**Comment 4**

Additionally, there is a need for accurate and comprehensive citation of all data sources in Table S1. Currently, it seems that only three references from the authors of this paper are cited. Ensuring that all data sources are correctly and comprehensively cited is crucial for maintaining the integrity of the research and providing clear, traceable scientific evidence.

*Response*

We have generated a comprehensive data source for all samples, which includes longitude, latitude, geographic location, habitats, project ID, and sample ID as a supplementary table.

Please see Table S1 and S2 in the attached files.

**Reviewer 3**

We would like to express our sincere gratitude to you for the immense dedication and time invested in this article, as well as for providing many valuable suggestions, which have greatly contributed to the quality of the manuscript.

**Comment 1**

Formatting:

Some headers contain letters that aren't bolded.

*Response:*

We have checked the format of the entire manuscript, and ensured that all text has been formatted consistently

**Comment 2**

Table S2 has cited data from Trivedi et al. in the future.

*Response*

We are sorry for the mistake due to the misused auto-filling function of Excel. It has been corrected as Trivedi et al., 2020.

**Comment 3**

Figure 1; The Alps are not considered the Arctic. Is the microbial richness calculated with all of the taxonomic identifications from metagenomes, cultures and amplicons?

*Response*

We have regrouped the samples, now the samples were grouped as Antarctic, Arctic, Tibetan Plateau, and other non-Polar glaciers.

**Comment 4**

In sentence, "Metagenomic assemblies were binned using MetaBAT 2 (v2.12.1) (Kang et al., 2019), MaxBin 2 (v2.2.7) (Wu et al., 2016), and VAMB (v2.0.1) (Nissen et al., 2021) respectively" respectively should be replaced with separately.

*Response*

We have changed the sentence as the reviewer suggested.

*Amended manuscript*

Metagenomic assemblies were binned using MetaBAT 2 (v2.12.1) (Kang et al., 2019), MaxBin 2 (v2.2.7) (Wu et al., 2016), and VAMB (v2.0.1) (Nissen et al., 2021) separately.

**Comment 5**

In "These MAG together with the downloaded isolate genomes were dereplicated using the thresholds of 30% aligned fraction and a genome-wide average nucleotide identity (ANI) threshold of 95% …" makes it sound like the MAGs and the cultured genomes

were dereplicated against each other instead of the method being applied to both sets of data, which I think is what is supposed to be conveyed.

*Response*

We have rephrased the sentence for clarity.

*Amended manuscript*

The obtained MAGs were combined with the isolate genomes, and these genomes were dereplicated using the thresholds of 10% aligned fraction and a genome-wide average nucleotide identity (ANI) threshold of 95%.

**Comment 6**

In "Spatially, 69.7% of all samples (n = 568) were from Tibetan glaciers, 24% (n = 196) were from Antarctic glaciers, while those from Arctic glaciers were slightly under-represented (6.3%, n = 51)." The arctic being slightly under-represented in an under statement.

*Response*

We have re-run the SRA entry filtering process and greatly improved the number of amplicon sequencing retrieved from the SRA dataset. Additionally, we have updated our data analysis pipeline, to allow a more samples can be kept after quality filtering. Please see attached Table 1 for details.

Thus, the Arctic data are no longer underrepresented

**Comment 7**

A main table displaying the number of samples in the cryoconite (sediment and water), ice and snow of the Arctic, Antarctica and Tibetan Plateau would help understanding.

*Response*

We have added a table in the main text to clarify the number of samples in the dataset. Please see attached Table 1 for details.

**Comment 8**

These sentences could be combined and refined for clarity. "We identified ubiquitous phylotypes for each region-habitat pair (i.e., identified in more than 55% of samples). There were five phylotypes identified as ubiquitous in all region-habitat pairs (Table S8), affiliated with Gammaproteobacteria (Comamonadaceae) or Actinobacteria (Microbacteriaceae).

*Response*

We rephrased the sentence according to the new results.

*Amended manuscript*

We then attempted to identify ubiquitous phylotypes for each region-habitat pair, defined as those present in more than 55% of samples with a relative abundance greater than 0.1%. However, we found no phylotypes that were common across any

regions or habitats (**Table S11**). The number of ubiquitous phylotypes varied, ranging from nine in Arctic ice to 129 in Tibetan-supraglacial meltwater. Notably, we did not identify any ubiquitous phylotypes in Arctic-snow or other non-polar-snow. Most of the ubiquitous phylotypes were associated with Gammaproteobacteria (29% of the total), followed by Bacteroidetes (19%), Alphaproteobacteria (16%), Cyanobacteria (13%), and Actinobacteria (11%). This distribution may indicate their ability to disperse and adapt to different environmental conditions.

**Comment 9**

Section 3.1.4 on Potential pathogens. Details are needed on what constitutes a pathogenic organism that made up this curated database.

*Response*

We have removed the potential pathogen sections due to the read length of amplicon sequencing does not meet the requirement for species-level classification

**Comment 10**

Line 195 – 196, "We propose that this could be explained by the similar selection mechanisms for long-distance dispersal survival and host-immune evasion" Is a bold claim without reporting how many of these phylotypes have the genes for the teichoic acid or reporting how many of these snow and ice pathogens were staphylococcus.

*Response*

The pathogen prediction part has been removed.

**Comment 11**

Line 203 "Of these dereplicated ORFs, 47.8% (29,947,128) were functional annotations using eggnog" should be 'functionally annotated'.

*Response*

We have corrected the sentence

*Amended manuscript*

Of these dereplicated ORFs, 47.8% (29,947,128) were functionally annotated using eggNOG.

**Comment 12**

Line 212, "…likely representing novel species" this cannot be claimed from gOTUs.

*Response*

We have rephrased the sentence as "Notably, 87% of the gOTUs were unable to be classified at the species level (**Fig. S5**), reflecting the genomic novelty of glacial microbiome."

**Comment 13**

Line 213-214. This is the first time South American glaciers are being considered alone.

*Response*

We have now classified as South American glaciers as other non-polar glaciers.

**Comment 14**

Figure 4b is difficult to read and discern and determine what habitat they come from. 4d would be improved by adding the locations were each of these gOTUs are present and if they relate to the organisms (phyla level) to the ones in figure 2b. 4e is written as 4f in the caption, and is difficult to apply these virulence factors to the

*Response*

We have modified the Fig. 4B, so that the number of genes from each habitats are displayed. For Fig. 4d, it is difficult to separate the genes by habitat as most of these genes are recovered from cultured isolated, thus the habitat information is not available. For Fig. 4e, we have added the taxonomy information for each gene.

Please see attached Fig. 4 for details.

**Comment 15**

Line 230 "relatively rare" add the percentage that these nitrogen fixation and nitrification genes make up.

*Response*

We have added the percentage of these genes relative to all nitrogen cycling genes identified

*Amended manuscript*

In comparison, genes involved in nitrogen fixation (*nifH*) and nitrification (*hao*) were relatively rare, with only 678 (0.49% of the nitrogen cycling-related genes) and 240 (0.17%) unique genes identified, respectively.

**Comment 16**

The Antimicrobial resistance genes section is fascinating and a great addition to the study.

*Response*

We thank you for acknowledging the values of this result

**Comment 17**

Line 268; The 4GDB link in inaccessible.

*Response*

We have changed the provider of the website, with a new address of https://nmdc.cn/4gdb/, the website is fully functional.

**Reviewer 4**

This manuscript proposes a dataset for supraglacial prokaryotic communities from glaciers in the Arctic, Antarctica, and from the Tibetan Plateau. It comprises amplicon sequence data, cultured bacterial genome data, and metagenomic data from the three main supraglacial habitats: snow, ice and cryoconite holes. The authors' research reveals a higher prokaryotic diversity on glaciers from Tibet in comparison to Arctic and Antarctic glaciers. Furthermore, the study of potential pathogens could help identify potential biohazards in glacial communities. This dataset is the first step in offering a study setting for a worldwide view of the prokaryotic community composition for supraglacial environments and covers microbial diversity, taxonomy, community structure, and genetic functions in glacial environments.

*Response*

Thank you for your thoughtful and constructive feedback on our manuscript, we appreciate the time and effort you dedicated to reviewing our work. We are grateful for your positive comments highlighting the significance of our dataset and its potential to provide a worldwide perspective on the prokaryotic community composition in supraglacial environments. We have responded your comments in a point-by-point style, please see our response below.

**Comment 1**

However, major concerns can be raised regarding the completeness, comparability and exploitation of the dataset. Among them, the lack of proper sourcing and credit attribution for data used but not produced by the authors. Table S1 should be completed replacing "NCBI SRA database" by proper credit when possible, including Millar et al., 2021, Webster-Brown et al., 2015, and others. Some samples, such as s2016ZPGSN, cannot be found on NCBI.

*Response*

We have added the manuscript DOI and author names wherever possible, otherwise they are cited as "unpublished data".

Please see table S1 and S2 in the attached file for details.

**Comment 2**

Driving conclusions on the bacterial composition and diversity from glacier samples all over the world, comparing studies performed over several years, seems risky without addressing and assessing first key points such as the use of different DNA extraction pathways, the evolution of sequencing techniques and the depth of sequencing they offer, and the different sequencing primer pairs used. The lack of proper credit to the authors and producers of the data retrieved from NCBI complicates further the access of the reader to such discrepancies in the assembled dataset.

*Response*

We agree that different primers, sequencing platform, depth, and strategy will influence the sequencing results. We have provided this information in the supplementary table. Please see attached Table S2 for details.

**Comment 3**

Furthermore, ~70% of the samples presented in this study come from the Tibetan plateau, inducing a considerable bias in the estimation of the bacterial diversity worldwide. PERMANOVA tests should be paired with beta dispersion tests to start addressing this issue. For a report on the three Poles, the under-representation in Arctic samples is concerning for the interpretation of this study's results.

*Response*

We have revised the samples included in the study, the sample number bias is now minimized. We also performed the permdisp analysis, significant differences in the distance to the centroid were found among most regions (except between Non-polar and Tibetan glaciers). Therefore, within group variations significantly influenced the between group community similarity.

*Amended manuscript*

PERMDISP analysis showed that snow samples exhibited a significantly (at the threshold of $P = 0.05$) higher dispersal from centroid (67.4%) compared to cryoconite (65.1%). Comparatively, those of ice core and algae were lower (57.9% and 58.9%, respectively), possibly due to the low sample numbers.

**Comment 4**

L98: Could you provide rarefaction curves justifying the use of this threshold?

*Response*

We have removed the pathogen prediction part, as this method could be unreliable.

**Comment 5**

L148-150: a higher diversity found in Tibetan cryoconite holes compared to snow and ice could be due to a sampling bias as well (which proportion of the Tibetan samples are from cryoconite holes?), which should be addressed.

*Response*

We agree that sampling bias heavily impact our results. In the revised manuscript, we refined the dataset included in the dataset and revised the analysis methods. The results dataset is much more balanced in sample number.

Please see table S1 in the attached files for details.

Minor revisions:

**Comment 6**

L15-16: as the authors study the prokaryotic community, it would be good to stick with the terms "bacterial and archaeal" or "prokaryotic" instead of "microbial" over the course of this manuscript.

*Response*

We have replaced microbial to "prokaryotic" where appropriate.

**Comment 7**

L22: "which could be attributed to the similar adaptation mechanisms for microbial survival in aerosol and immune evasion" this seems like a far-fetched conclusion.

*Response*

We have removed the pathogen prediction results, due to its inaccuracy.

**Comment 8**

L35-36: to reformulate. 6 petagrams (Pg) is the total organic carbon contained in glacier ice, and not all is released during the yearly seasonal melting.

*Response*

We have reformatted the sentence as "It has been estimated that six Pg of carbon are stored in global glaciers. These carbon may be released into downstream ecosystems with glacier runoff (Hood et al., 2015), influencing key elemental cycling in downstream ecosystems."

**Comment 9**

L41-43: grammatical changes needed

*Response*

We have rephrased the sentence as

"As microorganisms are the key driver of carbon and nitrogen transformation in glacier ecosystems, knowledge of their biogeography and functions can greatly enhance our understanding of the biogeochemical cycling in glacial ecosystems and aid in predicting the impact of climate change."

**Comment 10**

L49-50: citation needed

*Response*

We added appropriate citation for this statement

*Amended manuscript*

Algae and Cyanobacteria are the primary producers in supraglacial ecosystems, with other heterotrophic microorganisms participating in the transformation and degradation of endogenous and exogenous nutrients (Hotaling et al., 2017, Anesio et al., 2017).

**Comment 11**

L55-56: what is the justification for the mean microbial abundance in surface meltwater to increase with enhanced glacier retreat?

*Response*

We have added additional citation for this statement. In Segawa et al., 2005, the authors investigated the season variations in microbial biomass and found that the cell number of bacteria significantly increased during the melting season.

*Amended manuscript*

It was estimated that the mean microbial abundance in glacier surface meltwater is $10^4$ cells mL$^{-1}$ (Stevens et al., 2022), this quantity may further increase with the enhanced climate warming (Segawa et al., 2005).

**Comment 12**

L57-58: "which are not commonly monitored in the environment but have the potential to enter the environment" this needs enhanced clarity

*Response*

We have removed this sentence, as this is clearly redundant and only causes ambiguity.

**Comment 13**

L55-61: this part would benefit from a re-writing linking clearly the different findings to each other, to knowledge gaps and to the authors' hypotheses.

*Response*

We thank you for the kind suggestion. Nevertheless, this work presents a collection of data, and it is not aimed to answer any specific question(s). Therefore, we did not provide any hypotheses for this work. The section you mention is to briefly introduce what this dataset could be used for.

**Comment 14**

L69-70: archiving biological data from endangered environments such as glacier ecosystems is invaluable. However (and unfortunately), qualifying it of method allowing for the preservation of biodiversity is highly debatable.

*Response*

We have revised the sentences without stressing that our work could preserve biodiversity.

*Amended manuscript*

The dataset archives glacial-specific microorganisms and unique genes in digital form, thus representing an invaluable resource for bioprospecting.

**Comment 15**

L267: the link provided is not accessible

*Response*

We have changed the provider of the website, with a new address of https://nmdc.cn/4gdb/, the website is fully functional.

---

## Author Response (AR2)

**Comments from the reviewer**

General assessment

This manuscript presents a valuable contribution by assembling a large-scale, multi-source dataset of glacier microbiomes across the Three Poles and beyond. The integration of 16S rRNA amplicon data, metagenome-assembled genomes (MAGs), and cultured isolates is a strength, and the public database (4GDB) offers potential for long-term impact. However, several issues should be addressed before the manuscript can be considered for publication.

**1. Definition and scope of "microbiomes"**

The term "microbiomes" is generally understood to include a broad range of microbial life, including bacteria, archaea, microeukaryotes (e.g., algae and fungi), and viruses. While the manuscript refers to some of these groups in the introduction, the actual dataset and analyses are limited entirely to prokaryotes. There are no sequences or taxonomic assignments for eukaryotes or viruses.

I recommend clarifying this taxonomic scope explicitly in both the abstract and introduction. The current title may give the impression of broader taxonomic coverage than is actually presented, and I suggest revising it to reflect the prokaryote-specific focus more accurately.

**Response**

Thank you for the comments. We have specified that the dataset is "on supraglacial bacterial and archaeal (referred to as prokaryotic hereafter) communities across the Antarctic, Arctic, Tibetan Plateau, and other alpine regions". To further clarify the scope of our database, we changed the title of the manuscript to "A database of glacier prokaryotic genomes and genes for the Three Poles" and made other modifications listed below.

*Amended manuscript*

**Abstract**

Microbes, including bacteria, fungi, algae, and other microeukaryotes, are the primary inhabitants of glacier ecosystems and are key drivers of carbon and nitrogen transformation. Among them prokaryotes (including bacteria and archaea) are the most diverse and abundant.

The dataset contains 64,510 prokaryotic amplicon sequencing phylotypes, with a higher prokaryotic diversity in the Tibetan glaciers than in the Antarctic and Arctic glaciers…

The data can be leveraged to elucidate ecological principles governing the distribution of prokaryotes, to gain insights into the key functional genes for supraglacial microbiomes.

**Introduction**

From an ecological perspective, this dataset with standardized prokaryotic diversity, taxonomy, and community structure can improve understanding of the ecological principles governing the distribution of microorganisms across glaciers.

**2. Functional gene analysis**

The manuscript compiles an impressive array of functional gene annotations across thousands of MAGs, including pathways related to carbon degradation (CAZy), nitrogen cycling, methane metabolism, and antimicrobial resistance. However, the presentation remains largely descriptive.

While the authors provide a detailed catalog of gene abundances and distributions, the data are mostly reported in isolation, with limited interpretation in ecological or biogeochemical context. For example, the distinction between mmoX and pmoA gene distributions in cryoconite versus ice is noted, but the possible drivers—such as redox conditions, organic content, or microbial niche structure—are not explored. Similarly, CAZy profiles are summarized without considering variation across glacier types or habitats, and nitrogen-related genes are not discussed in terms of environmental gradients or host taxa.

Given the scope of the dataset, even basic exploratory analyses (e.g., ordination, correlation with metadata) could yield valuable insights. As it stands, this section reads more like a gene inventory than a functional synthesis.

**Response**

In response to your suggestion regarding expanding the ecological and biogeochemical interpretation of the gene distributions, we have incorporated new analyses and discussions to better link gene profiles with habitat differences and potential environmental drivers. Specifically, we added comparisons of CAZy gene distributions across different glacier types and habitats, explored the variations in nitrogen cycling genes in relation to habitat-specific microbial taxa and conditions, and discussed the environmental significance of methane monooxygenase gene distributions in cryoconite versus ice.

However, we would like to kindly note that the primary aim of our manuscript is to provide a comprehensive, original dataset and a functional gene catalogue across glacier-associated habitats. Earth System Science Data focuses on the dissemination and sharing of data resources that can support further Earth system research. While we agree that more detailed mechanistic or process-based ecological analyses would enrich interpretations, such in-depth studies are beyond the intended data reporting and synthesis scope of the manuscript.

*Amended manuscript*

**For the comment regarding that CAZy profiles are summarized without considering variation across glacier types or habitats**

The dataset contains 1,082,125 genes encoding carbohydrate-active enzymes (CAZymes, **Fig. 5a**), i.e., those enzymes involved in the metabolism of glycoconjugates, oligosaccharides, and polysaccharides (Zerillo et al., 2013). Genes associated with carbohydrate hydrolysis (GH) and biosynthesis (GT) were the most abundant, accounting for 45.2% and 44.4%, respectively. In contrast, those genes associated with non-hydrolytic cleavage of glycosidic bonds (PL), hydrolysis of carbohydrate esters (CEW), and assisting in degrading biomass substrates (AA) were relatively scarce, accounting for 0.8%, 3.1%, and 0.2% of the predicted CAZY, respectively. This indicates that the glacier microbiome is competent in a diverse range of carbon

transformation processes, mediating the delivery of carbon to downstream ecosystems. We further examined the distribution of CAZyme genes across different glaciers. At the CAZY Family level, GT2, GT4, GH13, and GT51 are the most diverse across all habitats (**Fig. 5b**). PERMANOVA tests revealed significant composition differences by both habitat (pseudo-F = 10.014, $P < 0.001$) and regions (pseudo-F = 5.038, $P = 0.002$), with habitat exhibiting a greater influence on CAZY composition. Specifically, the PCA plot revealed a greater number of genes classified as GH5 and GH9 in cryoconites (**Fig. 5c**). Furthermore, 30 GHs and 20 GTs exhibited significantly higher contribution in cryoconites than in ice or snow (**Fig. 5d**). Comparatively, 15 GHs and 27 GTs exhibited significantly higher contribution in ice or snow than in cryoconites. Thus, cryoconites exhibited higher capacity in the catabolism of organic carbon, whereas snow and ice are dominated by anabolism.

[Figure]

**Fig. 5 The distribution of genes associated with carbohydrate-active enzymes (CAZymes) in glaciers.**

a: The relative contributions of different genes associated with CAZymes in each habitat (GH: Glycoside hydrolases; GT: glycosyl transferases; PL: Polysaccharide lyases; CE: Carbohydrate esterases; AA: Auxiliary activities); b: The most diverse CAZymers in each habitat-region combination. c: PCA plots shows the distribution of CAZymes across habitats and regions; d: the selective enrichment of CAZymes between cryoconite and snow and between cryoconite and ice.

**For the comment regarding that Nitrogen-related genes are not discussed in terms of environmental gradients or host taxa.**

The differences in gene distributions are apparent among different habitats. Specifically, the genes associated with nitrogen fixation and denitrification are mainly present in cryoconites. The former could be explained by the high abundance of Cyanobacteria (**Fig. 6**), while the latter could be attributed to its anaerobic conditions. Thus, the

cryoconite could be a potential source of $N_2O$. Comparatively, *amoA* (in nitrification) is mainly identified from ice, while *norB* and *nosZ* (in denitrification) are also abundant. This reflects that ice could harbor greater functional diversity than snow, with the capacity for nitrogen transformation and influencing the nutrient discharged to downstream ecosystems.

**For the comment regarding the Distinction between mmoX and pmoA gene distributions in cryoconite versus ice**

The two forms of methane monooxygenase (sMMO and pMMO) have distinct enzymatic characteristics and are expressed in different growth conditions. Specifically, sMMO is expressed under low copper conditions ($\leq 0.9$ nmol of Cu/mg of cell protein), while pMMO is expressed under relatively high copper/biomass ratios (Zhang et al., 2017). Furthermore, sMMO has a broader substrate range, can oxidize methane, short chain alkane, alkene, and aromatic compounds, while pMMO can only oxidize alkanes of less than five carbons (Trotsenko and Murrell, 2008). Despite being not empirically measured, the cryoconite is expected to contain a higher concentration of copper than the ice, as the former is precipitated dust. Thus, the higher diversity of sMMO in the cryoconite could be associated with its ability to oxidize diverse alkanes, which may be present in soils. In comparison, high-affinity pMMO could oxidize methane at atmospheric levels, which may support microbial communities in the oligotrophic ice habitat. Only six unique methanogenesis-related genes (*mcrA*) were identified, almost exclusively in cryoconite metagenomes. This is consistent with the cryoconite as a methane source in the literature (Zhang et al., 2021).

**3. Amplicon analysis**

The alpha and beta diversity analyses are competently executed, and the dataset is extensive. However, the manuscript does not explain **how differences in sequencing protocols**, **depth**, or **metadata completeness** across studies were handled.

Some discussion of **data harmonization strategies**, **rarefaction**, or the **potential for batch effects** would be helpful—particularly because inter-study comparisons are a central part of the analysis.

**Response**

Data homogenization is important for comparison among different datasets. We further discussed the harmonization strategy used, assessed the sequencing depth, and evaluated the influence of batch effect and primer usage. Specifically, we performed Kruskal Kruskal-Wallis one-way ANOVA test on all alpha diversity indices by the projects (for batch effects), the primers used, the amplified regions, and the sequencing platforms. The results showed that most indices are significantly different by these factors. However, this could be due to the different study areas employed. Thus, we repeated the comparison among different habitats and regions using data that was generated using the same primer set. The result patterns are consistent with our analysis using the entire dataset. Thus, the selection does not impact the conclusions. Similarly, we performed PERMANVOA analysis on the community structure, with consistent result patterns being observed, indicating that the different primers and studies do not affect the validity of the results.

We have added additional results and discussions to clarify this and commented on the completeness of metadata and sequencing depth.

*Amended manuscript*

**For the discussions on the harmonization strategy**

A variety of primers were used by these projects, amplifying the hypervariable regions V3V4, V4, and V4V5 regions. These data were harmonized by retaining only the V4 region (sequence trimming). Surprisingly, four bioprojects that used primers 783F and 1046R (V5V6 region) were also retained. We speculate that incorrect primers may have been provided in the NCBI. Nevertheless, we provided necessary information on the primer and regions amplified, users may choose to use the entire dataset or only those using the same primer or same amplification region.

**Comments on metadata availability**

The retained datasets are originated from 66 bioprojects, eight of which missed sequencing platform information and two of which missed primer information (Table S2). Most of these bioprojects do not have environmental metadata, therefore are not included in the dataset.

**Assessments on sequencing depth**

The Good's coverage index provides estimation for the number of singletons in a sample, reflects the coverage of the sequencing. The values of the index were 0.98±0.02 and 0.96±0.02 for the datasets without and with subsampling, respectively (Table S7). This indicates that majority of the OTUs were identified.

**Evaluation on the batch effect and primer usage on alpha diversity indices**

We further assessed the influence of the region amplified on the validity of the results. For each region-habitat pair, the alpha diversity indices were significantly different by these factors to a certain extent. However, those from Arctic basal ice, non-polar glacier ice, and Tibetan Plateau supraglacial meltwater were less affected (**Table S9**). Nevertheless, the influence may be explained by the different sampling locations, which have distinct microbial composition. We further tested the validity of the diversity comparison results using the data that were generated using the same primer set (**Table S10**). The influence of primer selection on prokaryotic diversity was inconsistent. For instance, primers targeting the V4 region resulted in a higher richness in supraglacial ice than primers targeting the V3V4 region in other alpine glaciers. In contrast, the primers targeting the V3V4 region had a higher in the Arctic. Such inconsistency in microbial community assessment by different primers and platforms has been reported previously (Fredriksson et al., 2013; Tremblay et al., 2015). Thus, the homogenization method may not fully overcome the bias caused by primer selection. Nevertheless, we provided necessary information on the primer and regions amplified, user of the dataset may choose to use the entire dataset or only those using the same primer for analysis.

**4. Database usability and FAIR principles**

The 4GDB appears to be a useful and well-organized resource. However, the manuscript provides little detail regarding its long-term maintenance or accessibility features.

It remains unclear how often the database will be updated, whether APIs or bulk downloads are available, or under what license the data can be reused. A short section outlining how 4GDB aligns with FAIR principles—especially regarding reuse and interoperability—would enhance transparency and user trust.

**Response**

The website is constructed under the OSI-approved CC BY 4.0 Open Source license (https://creativecommons.org/licenses/by/4.0/), all data can be accessed and reused freely, without any restrictions, for both academic and commercial purposes. Regarding the FAIR (Findable, Accessible, Interoperable and Reusable) principles. We permit bulk download at https://nmdc.cn/4gdb/download, which provide compressed files for the genomes of the recovered MAGs, predicted genes from the MAGs, and the OTU tables for the 16S rRNA gene amplicon sequencing. These data can be downloaded using FTP client such as FileZilla. For the raw sequencing files, they are stored in NCBI and EMBL-EBI, we do not provide direct download link for these files, but listed the accession number to redirect the user to the correct data. For predicted genes from the metagenome, we provide blast search option, so that the user can find the genes of interest and download them individually. Due to the large size, we don't provide bulk download for these sequences. The website will be maintained by the author teams, at the stage, we will try to update at least once per year.

*Amended manuscript*

The website is constructed Under the OSI-approved CC BY 4.0 Open Source license (https://creativecommons.org/licenses/by/4.0/), all data can be accessed and reused freely, without any restrictions, for both academic and commercial purposes.

The 4GDB website is mainly structured into three sections, comprising amplicon sequencing, metagenome/genome sequences, and function prediction. The user-friendly web interface allows data filtering based on sample type, sample location, habitat type, gene type, and taxonomy, enabling seamless download of the filtered results. In conclusion, 4GDB (https://nmdc.cn/4gdb/downloadtemp) provides an open-access genome- and gene-orientated resource platform that is regularly updated to include newly published and in-house generated sequence data.